# Predicting dominant terrestrial biomes at a global scale using machine learning algorithms, climate variable indices, and extreme event indices

Hisashi Sato [1,2]*

1  Research Institute for Global Change, Japan Agency for Marine-Earth Science and Technology (JAMSTEC), Yokohama, Japan, 2  Department of Ecosystem Studies, Graduate School of Agricultural and Life Sciences, The University of Tokyo, Tokyo, Japan

* hsatoscb@gmail.com

## Abstract

Understanding the global distribution of biomes is essential for biodiversity conservation, climate modeling, and land-use planning. Traditional approaches often summarize climate data into indices, and recent models sometimes include extreme events such as severe droughts or rare cold spells. This study evaluates how the choice of machine learning algorithm, climate data summarization, and extreme climate indices affect the accuracy and robustness of global biome modeling. Four algorithms were tested: random forest (RF), support vector machine (SVM), naive Bayes (NV), and LeNet convolutional neural network (CNN). RF and CNN achieved the highest accuracy, with CNN preferred due to RF's stronger overfitting. Summarizing climate data into indices reduced accuracy by 1–2%, while adding extreme indices increased accuracy by <2% (except for NV, which performed poorly overall). However, extreme climate data caused large mismatches between observed and predicted climate values, reducing robustness as measured by prediction consistency. These results indicate that including extreme climate data in global biome prediction models offers limited accuracy gains but can significantly weaken robustness, so caution is advised.

## Introduction

Biomes—major regional ecological community defined by distinctive life forms and dominant plant species [1]—are primarily determined by climate [2,3] and, in turn, influence climate through biophysical and biochemical feedbacks [4]. Understanding biome distributions is therefore essential not only for estimating land potential and guiding conservation policy, but also for informing climate projection models (reviewed in Hengl, Walsh [5]).

Many methods have been proposed to model biome distributions (reviewed by Sato and Ise [6]). Traditional approaches often use a limited set of derived climate

**Data availability statement:** All data required to reproduce the analyses described herein are openly available via Zenodo (https://doi.org/10.5281/zenodo.8113935).

**Funding:** HS received two grants from the Ministry of Education, Culture, Sports, Science and Technology (MEXT), Japan: [[Arctic Challenge for Sustainability II (ArCS II) [Program Grant Number JPMXD1420318865] [Arctic Challenge for Sustainability III (ArCS III) [Program Grant Number JPMXD1720251001]]. The funders had no role in study design, data collection and analysis, decision to publish, or preparation of the manuscript.

**Competing interests:** The authors have declared that no competing interests exist.

indices (e.g., annual precipitation, coldest-month temperature) to summarize monthly or seasonal data. However, recent advances in machine learning (ML) allow models to incorporate a larger number of raw variables without summarization, potentially improving accuracy and flexibility. For example, Hengl, Walsh [5] used 160 environmental variables, including soil and topography, and monthly climate variables, to model biome distribution using ML algorithms. However, excessive input dimensionality may reduce generalizability and increase computational cost, underscoring the need to balance model complexity and performance. Among the diverse variables now accessible through ML methods, extreme climate events—such as severe droughts and rare low-temperature incidents—have emerged as particularly important predictors of boundaries of biomes (reviewed in Beigaite, Tang [7]). Incorporating extreme climate indices alongside regular variables has been found to improve the performance of decision tree-based models [7].

This study evaluates the effectiveness of machine learning models in predicting the global distribution of potential natural vegetation (PNV), i.e., vegetation in the absence of human influence, under current and future climate conditions. It addresses three questions: (1) Which machine learning algorithm performs best in reproducing the current biome distribution? (2) Does using raw monthly climate variables improve model performance compared to using derived climate indices (BIOCLIM)? (3) Does incorporating extreme climate indices (CLIMDEX) improve model robustness under projected future conditions (2060–2080, RCP8.5)? The analysis focuses on prediction accuracy, model stability, and spatial consistency under extrapolative scenarios.

To contextualize the modeling approach adopted in this study, two reference points are particularly relevant. First, process-based dynamic global vegetation models (DGVMs) simulate vegetation dynamics through mechanistic representations of ecosystem processes [8]. These include plant physiological functions, carbon cycling, and interspecies competition. While this allows for detailed representation of ecological mechanisms, DGVMs rely on numerous parameterizations and assumptions, which can introduce uncertainty—especially when projecting into novel future climates. In contrast, this study, like the one by [9], adopts a purely data-driven approach using ensemble machine learning algorithms that infer climate–vegetation relationships directly from observed data. This enables the models to capture empirical patterns with fewer structural assumptions, offering a computationally efficient and flexible alternative for large-scale biome prediction, particularly under extrapolative scenarios. Second, unlike land cover maps derived from satellite imagery, which primarily reflect current human-modified landscapes, this study aims to model potential natural vegetation (PNV)—that is, the vegetation that would exist without anthropogenic disturbance—at a 0.5° resolution. In this context, data-driven modeling provides an effective means for identifying climate-sensitive biome boundaries and anticipating potential shifts under future scenarios, independent of current and future land-use patterns.

## Methods

### Biome data

This study used the potential natural vegetation (PNV) dataset compiled by Beigaite, Tang (7) for their decision tree-based models of the global PNV distribution. Here, the same dataset is applied to a wider range of machine learning algorithms. In their study, the authors derived the PNV data from the Moderate Resolution Imaging Spectroradiometer (MODIS) MCD12C1 land cover product in 2001 [10], specifically using the International Geosphere Biosphere Programme (IGBP) land cover classification. This dataset based on the supervised learning classification of the MODIS Terra and Aqua reflectance data [11], contains percent cover for 17 IGBP classes [12] in each grid cell at a resolution of 0.05°. They resampled the data to 50 km × 50 km grids and identified the dominant natural vegetation type in each grid cell. Grid cells partly affected by human activity were retained, under the assumption that relative proportions of natural vegetation remain stable despite such modifications. Of the original 17 categories, only 13 representing natural vegetation were used (Fig 1, S1 Table). Cells with 100% human activity, water cover, or both were excluded as was Antarctica, leaving 52,297 grid cells for analysis. Although Beigaite, Tang (7) did not specify the proportion removed, it is estimated that roughly 10–11% of the land grid cells (excluding Antarctica) were excluded.

Although many biome classifications exist (e.g., [13,14]), I used this PNV dataset for three practical reasons aligned with global climate-vegetation modelling at 0.5°: (i) the workflow-deriving the dominant natural class from MODIS/IGBP and resampling to ~50-km grids-matches my analysis grid; (ii) its climate inputs and projections are processed consistently with the BIOCLIM (*Avel*) and CLIMDEX (*CEI*) indices used here, with harmonized definitions and resolution for present data and CMIP5-based futures; and (iii) employing the same dataset as related machine-learning studies facilitates comparability without additional preprocessing.

Because the aim of this study is to model potential natural vegetation (PNV), I retained grid cells that are only partially affected by human activity, on the premise that the dominant natural-vegetation signal remains informative at 0.5°

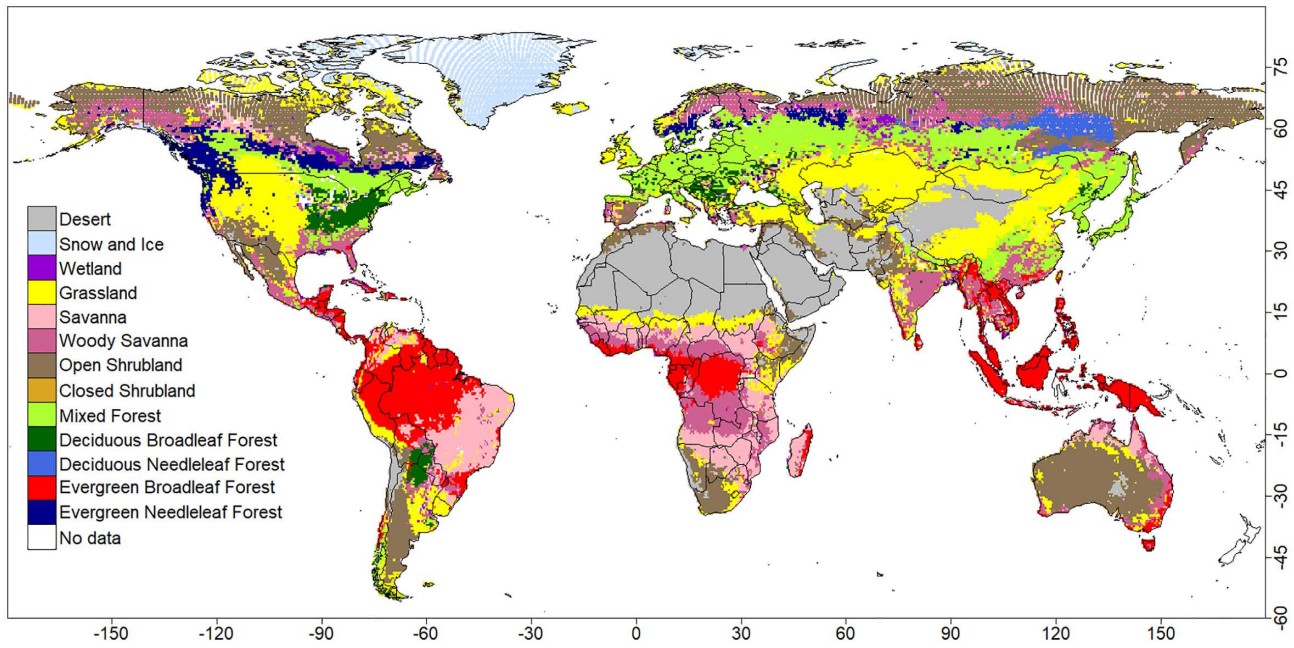

**Fig 1. Distribution of observation-based PNV data, which were used to train machine learning-based models in this study.**

resolution. By contrast, cells with 100% human cover and/or water were excluded, leaving 52,297 land grid cells (approximately 10–11% of land cells excluded, Antarctica removed). This choice minimizes spatial coverage bias while preserving information on the prevailing natural type. I acknowledge, however, that uncertainty in PNV estimation may increase in regions with strong human influence, and I interpret results in those areas with caution.

## Machine learning algorithms

Four machine learning algorithms were employed: RF [15], SVM [16], NV [17], and CNN [6]. RF, SVM, and NV were implemented in R v3.3.3 [18] using the randomForest, ksvm, and naiveBayes packages, respectively, with commands such as randomForest(*VegNo* ~ ., *DatasetTrain*), where *DatasetTrain* is the training dataset and *VegNo* is the biome category column. All models were run with default settings to (1) ensure fair comparability by avoiding bias from parameter tuning, (2) keep the implementation straightforward and reproducible, and (3) align with the study's objective of evaluating algorithms and data combinations rather than optimizing a single best-performing model. This choice also simplified implementation. The RF model, for instance, already achieved 100% accuracies on the training sets with default parameters, indicating strong overfitting and suggesting that further tuning would not improve generalization in this case. While strategies such as cross-validation and variable selection can reduce overfitting [19], the RF results indicate that high-capacity models may still overfit even under best-practices settings.

In contrast to Beigaite, Tang (7), a decision tree algorithm was not included in this study. Although decision trees rapidly provide interpretable boundary conditions for the distribution of a given output variable, they are generally inferior to the algorithms explored in this study in terms of reconstruction accuracy [20]. The RF algorithm is an ensemble of decision tree algorithms, which should provide higher model accuracy [15].

Although the models other than CNN were trained using numerical climate data, applying CNN algorithms requires converting the climate data into images. CNNs are typically applied to analyze visual imagery and have been successfully adapted for species distribution modeling at regional [21,22] and global scales [6]. The present CNN was trained following the method of Sato and Ise (6). This method represents climatic conditions using graphical images and employs them as training, testing, and prediction data for CNN models. I selected this method because it allows CNNs—originally developed for image analysis—to automatically extract nonlinear seasonal patterns from multiple climate variables while preserving their temporal structure, enabling convolutional filters to identify spatially coherent features. This method automatically extracts nonlinear seasonal patterns for climatic variables relevant to biome classification. Each graphical image is 256×256 pixels and is divided into rectangular cells representing each data point, with tiles in each cell expressing the values in grayscale. Prior to this visualization, climate variables were standardized to the range 0.01–1.00 using a log transformation. The R code for generating the images is available in the online open data repository.

In this paper, the term "CNN" is used, for convenience, to refer to the method of Sato and Ise (6), a machine-learning approach that includes transforming climate data into images as an integral part of its framework. In this method, the arbitrariness inherent in converting climate data into images can lead to variation in learning accuracies. However, Sato and Ise (6) demonstrated that, despite employing various strategies for image transformation and evaluating the resulting differences in learning performance, the effects on biome prediction were minimal: the range of training accuracies was 58.3–59.7% among four different imaging methods (such as pie charts and various color palettes), and 56.2–57.8% among four schemes used to transform climatic variables prior to graphical conversion (such as linear, log, and sigmoid transformations). Although these results are based on different climate and biome datasets than those used in the present study, they provide a useful reference.

The four algorithms selected in this study represent contrasting assumptions and capabilities in handling nonlinearity and feature interactions. RF is an ensemble-based decision tree method known for its robustness, but it can easily overfit training data without proper regularization or depth constraints, especially when using default settings [23]. CNNs have a strong capacity to extract complex nonlinear patterns from spatially structured inputs and have demonstrated superior

performance in recognizing hierarchical relationships in multidimensional data [24]. SVMs and NV represent more traditional machine learning approaches: SVMs are sensitive to data scaling and kernel choice, while NV classifiers rely on conditional independence assumptions that may not hold for ecological data. Including these diverse algorithms allowed a broad assessment of how model assumptions influence biome classification performance. Although CNNs were trained using visually encoded climate data rather than tabular variables, the underlying information was identical across all algorithms, ensuring comparability despite differences in input format.

## Climate data

This study used four climate datasets: averaged monthly air temperature and precipitation (*Ave*, 24 variables), averaged monthly climate indices (*AveI*, 16 variables), climate extreme indices representing extreme conditions on a daily scale such as the maximum length of a dry spell (*CEI*, 27 variables), and a subset of *CEI* (*CEI_{part}*, 21 variables). The variables included in *AveI* and *CEI* are listed in Tables 1 and 2, respectively. S1–S3 Figs show the present (1970–2000) and future (2061–2080) distributions of *Ave*, *AveI*, and *CEI*, respectively. Among all of the climatic variables used in this study, only six in the *CEI* dataset (Tn10p, Tx10p, Tn90p, Tx90p, WSDI, and CSDI) had completely separate distributions between the present and future. Another indexed extreme climate dataset, *CEI_{part}*, was constructed by excluding these variables from the *CEI* dataset.

The *Ave* data were obtained from the WorldClim version 2.1 product (released January 2020; Fick and Hijmans (25)), which represents average monthly air temperature and precipitation data for 1970–2000. The original WorldClim 2.1 product [25] was downloaded at a spatial resolution of 10 min, and resampled to 50 km × 50 km grids using the nearest-neighbor method. *AveI* was released by Beigaite, Tang (7), summarizing WorldClim 2.1 properties in terms of annual means (e.g., BIO1 and BIO12), seasonality (e.g., BIO4, BIO7, and BIO15), and limiting environmental factors to a monthly scale (e.g., BIO5, BIO6, and BIO14).

The *CEI* product was also released by Beigaite, Tang (7) using the CLIMDEX [26,27]. CLIMDEX comprises four datasets that were derived from different reanalysis datasets. Among these, Beigaite, Tang (7) used a dataset calculated

**Table 1. Indexed (summarized) monthly climate data (*AveI*).**

| ID | Description | Unit |
|---|---|---|
| Bio1 | Annual mean temperature | °C |
| Bio2 | Mean diurnal range (mean of monthly [range]) | °C |
| Bio3 | Isothermality | °C |
| Bio5 | Maximum temperature of the warmest month | °C |
| Bio6 | Minimum temperature of the coldest month | °C |
| Bio8 | Mean temperature of the wettest quarter | °C |
| Bio9 | Mean temperature of the driest quarter | °C |
| Bio10 | Mean temperature of the warmest quarter | °C |
| Bio11 | Mean temperature of the coldest quarter | °C |
| Bio12 | Annual precipitation | mm |
| Bio13 | Precipitation of the wettest month | mm |
| Bio14 | Precipitation of the driest month | mm |
| Bio16 | Precipitation of the wettest quarter | mm |
| Bio17 | Precipitation of the driest quarter | mm |
| Bio18 | Precipitation of the warmest quarter | mm |
| Bio19 | Precipitation of the coldest quarter | mm |

Note: Units: °C = degrees Celsius; mm = millimeters.

**Table 2. Indexed extreme climate (*CEI*).**

| ID | Description | Unit |
|---|---|---|
| FD | Number of frost days: annual count of days when TN (daily minimum) < 0°C | days |
| SU | Number of summer days: annual count of days when TX (daily maximum temperature) > 25°C | days |
| ID | Number of icing days: annual count of days when TX (daily maximum temperature) < 0°C | days |
| TR | Number of tropical nights: annual count of days when TN (daily minimum temperature) > 20°C | days |
| GSL | Growing season length: annual (January 1 to December 31 in the Northern Hemisphere (NH), July 1 to June 30 in the Southern Hemisphere (SH)) count between first span of at least 6 days with daily mean temperature > 5°C and first span after July 1 in NH (January 1 in SH) of 6 days with TG < 5°C | days |
| TXx | Monthly maximum value of daily maximum temperature | °C |
| TNx | Monthly maximum value of daily minimum temperature | °C |
| TXn | Monthly minimum value of daily maximum temperature | °C |
| TNn | Monthly minimum value of daily minimum temperature | °C |
| Tn10p * | Cool nights: percentage of days when TN < 10th percentile | % |
| Tx10p * | Cool days: percentage of days when TX < 10th percentile | % |
| Tn90p * | Warm nights: percentage of days when TN > 90th percentile | % |
| Tx90p * | Warm days: percentage of days when TX > 90th percentile | % |
| WSDI * | Warm spell duration index: annual count of days with at least 6 consecutive days when TX > 90th percentile | days |
| CSDI * | Cold spell duration index: annual count of days with at least 6 consecutive days when TN < 10th percentile | days |
| DTR | Diurnal temperature range: monthly mean value of the difference between Tx and Tn | °C |
| Rx1day | Monthly maximum consecutive 1-day precipitation | mm |
| Rx5day | Monthly maximum consecutive 5-day precipitation | mm |
| SDII | Simple precipitation intensity index: annual total precipitation divided by the number of wet days (defined as PRCP ≥ 1.0 mm) in the year | mm/day |
| R10mm | Number of heavy precipitation days: annual count of days when PRCP ≥ 10 mm | days |
| R20mm | Number of very heavy precipitation days: annual count of days when PRCP ≥ 20 mm | days |
| R1mm | Number of wet days: annual count of days when PRCP ≥ 1 mm (days) | days |
| CDD | Maximum length of dry spell: maximum number of consecutive days with RR (daily precipitation amount) < 1 mm | days |
| CWD | Maximum length of wet spell: maximum number of consecutive days with RR ≥ 1 mm | days |
| R95p | Very wet days precipitation: annual total PRCP when RR > 95th percentile | mm |
| R99p | Extremely wet days precipitation: annual total PRCP when RR > 99th percentile | mm |
| PRCP-TOT | Annual total precipitation on wet days (RR ≥ 1 mm) | mm |

* $CEI_{part}$ does not contain these six variables.

from the ERA-Interim reanalysis dataset, which accurately reproduces the observed climate extremes [28]. The *CEI* data derived from the ERA-Interim reanalysis dataset covers 32 years (1979–2010). Multi-year *CEI* values were averaged for each grid; multi-year averages of extreme indices are commonly used to represent average extreme conditions in the past and future [26,27,29]. The original resolution of the *CEI* data was 1.5° × 1.5°; they were transformed onto 10 min × 10 min grids through conservative interpolation and then resampled to 50 km × 50 km grids using nearest-neighbor interpolation.

Future climate conditions (2061–2080) were projected using BIOCLIM [25] and CLIMDEX [26,27] indices derived from the Intergovernmental Panel on Climate Change (IPCC) Coupled Model Intercomparison Project Phase 5 (CMIP5) ensemble means of 11 models, averaged over that period. RCP8.5 was selected as it represents the most severe climate change scenario, providing a stringent test of model robustness under conditions far beyond the present-day range. In the IPCC's Fifth Assessment Report (AR5, 2014), RCP8.5 assumes continued growth in global GHG emissions throughout the 21st century, reaching ~758 ppm $CO_2$ by 2080 [30]. All variables were standardized to ensure compatibility between present and future datasets. Present-day BIOCLIM indices were obtained directly from WorldClim v2 [25]. Future BIOCLIM indices were generated from CMIP5 outputs that had been bias-corrected and adjusted to match the definitions and resolution and definition of the present-day BIOCLIM data. CLIMDEX indices were calculated using the same methods for historical and projected data [26,27]. These data sources and processing steps follow the same approach used in [7]. Equivalent CMIP6 products with the same resolution and correction methods are not yet widely available.

Intermediate projection periods (e.g., 2041–2060) were not used because the aim was to test model robustness under the most extreme climate conditions. Since RCP8.5 represents the highest emission trajectory, its far-future projection (2061–2080) was considered sufficient for evaluating extrapolative performance.

## Data analysis

The learning performance of six climate dataset combinations—*Ave*, *Ave*+*CEI*, *Ave*+*CEI*$_{part}$, *AveI*, *AveI*+*CEI*, and *AveI*+*CEI*$_{part}$—was compared to disentangle three effects: (1) summarizing the climate data into indices, (2) adding extreme climate indices, and (3) adding overlapping extreme climate indices whose distributions are shared between current and future conditions. Four machine learning algorithms were applied to each dataset combination, resulting in 24 models in total. By structuring the analysis this way, I avoided the complexity of interpreting results from an exhaustive examination of which specific feature combinations the machine learning algorithms most strongly depended on.

For each model, 25% of all 52,297 grids (13,074 grids) were randomly selected for training, and the remaining 75% (39,223 grids) for testing. This proportion is the reverse of the typical 70–80% allocation to training [31], chosen to emphasize model robustness over performance and to ensure that rare vegetation types (<1%) were adequately represented in the test set. Training accuracy was defined as the proportion of correct predictions on the training data, and test accuracy as the proportion on the test data; their difference was used as the overfitting score [32]. To reduce sampling bias and better assess generalization, each model was evaluated in ten replicate experiments using different random seeds. The results were averaged across the ten independent 25:75 splits, serving a similar role to k-fold cross-validation.

To evaluate model robustness under future climate scenarios, I compared biome maps produced by different machine learning algorithms trained on the same dataset. Here, robustness refers to the stability of predictions across different algorithms. It was measured using the pairwise coincidence rate—the percentage of grid cells where two models assigned the same biome class. This metric captures agreement among models regardless of their match with the observed map, providing an indicator of consistency rather than accuracy (see Table 4).

## Results

### Overall model accuracy

Across all training datasets, three of the four machine learning models—RF, CNN, and SVM—showed high test accuracy in reconstructing global PNV distributions (Table 3). Test accuracy ranged from 80.1–81.4% for RF, 77.1–82.0% for CNN,

**Table 3. Test accuracy, training accuracy, and overfitting scores (% mean±standard deviation; n=10) for models based on four machine learning algorithms: random forest (RF), support vector machine (SVM), Naive Bayes classifier (NV), and convolutional neural network (CNN). Input variable abbreviations: *Ave*, averaged monthly air temperature and precipitation; *AveI*, averaged monthly climate indices; *CEI*, climate extreme indices; and *CEI*$_{part}$, a subset of *CEI*.**

| Input variable combinations | RF | SVM | NV | CNN |
|---|---|---|---|---|
| *Ave* | 81.2±0.21<br>100.0±0.00<br>18.9±0.21 | 76.4±0.15<br>77.8±0.28<br>1.38±0.30 | 46.7±0.84<br>46.8±0.70<br>0.11±0.40 | 79.1±0.15<br>81.1±0.90<br>2.05±0.99 |
| *Ave+CEI* | 81.4±0.20<br>100.0±0.00<br>18.7±0.20 | 78.0±0.15<br>79.9±0.25<br>1.92±0.30 | 45.2±1.20<br>45.3±1.02<br>0.15±0.36 | 80.1±0.12<br>81.9±0.94<br>1.78±0.91 |
| *Ave+CEI*$_{part}$ | 81.5±0.22<br>100.0±0.00<br>18.6±0.22 | 77.7±0.19<br>79.5±0.32<br>1.83±0.43 | 44.2±1.24<br>44.4±1.08<br>0.14±0.32 | 81.8±0.30<br>83.0±0.44<br>0.75±0.62 |
| *AveI* | 80.1±0.22<br>100.0±0.00<br>20.0±0.22 | 74.6±0.12<br>76.1±0.33<br>1.53±0.39 | 50.1±0.88<br>50.5±0.97<br>0.33±0.59 | 77.1±0.18<br>78.3±1.02<br>1.19±1.03 |
| *AveI+CEI* | 81.2±0.21<br>100.0±0.00<br>18.8±0.21 | 77.7±0.19<br>79.8±0.14<br>2.05±0.29 | 44.6±1.82<br>44.7±1.74<br>0.15±0.49 | 79.9±0.16<br>82.1±0.90<br>2.17±0.86 |
| *AveI+CEI*$_{part}$ | 81.3±0.18<br>100.0±0.00<br>18.8±0.18 | 76.9±0.16<br>78.5±0.38<br>1.91±0.46 | 43.3±2.26<br>43.5±2.14<br>0.16±0.43 | 82.0±0.31<br>82.9±0.48<br>1.06±0.57 |

74.6–78.0% for SVM, and 44.2–50.1% for NV. Based on these values, the ranking in descending order of accuracy was RF, CNN, SVM, and NV. The NV model performed markedly worse, with large misclassification areas in boreal and tropical forests (Figs 1, S4–S7 Fig).

All models exceeded the baseline accuracy of 17.8%, obtained by predicting all grid cells as the most frequent PNV class, grassland (SI S1 Table). As a further reference, a naive model assigning each grid cell the most frequent PNV observed at its latitude achieved 49% accuracy—lower than all machine learning models except NV.

The NV model's low test accuracy was mainly due to overestimating areas dominated by boreal forest, tropical rainforest, and deciduous broadleaf forest (Fig 4). In contract, the other models primarily showed discrepancies along PNV boundaries (Figs 2, 3, and 5), consistent with the observed fragmentation of biome distributions along PNV boundaries (Fig 1). The biome distributions reconstructed by the models, however, exhibited more continuous structures (Figs. S4–S7 Fig). The NV model's poor performance likely stem from its assumption that prediction variables are independent—a condition violated by climate variables. The NV model was excluded due to its poor performance from further analysis and discussion Fig 4.

## Error analysis

All models showed similar weaknesses in classifying certain small-area biomes, "Wetlands" and "Closed Shrublands" (0.5% and 0.2% of grid cells, respectively). Using the most basic *AVE* dataset, test accuracy for Wetlands was RF: 23.4%, SVM: 0.0%, and CNN: 3.6%. For Closed Shrublands, the corresponding values were RF: 7.3%, SVM: 29.0%, and CNN: 5.7% (S2-S4 Tables). Notably, the SVM model did not classify any grid cells as "Wetland" across all 39,233 grids×10 tests. These errors were widely dispersed and occurred in all models. The poor performance for these small-area biomes likely reflects their limited representation in the training data, which in turn reduces classification accuracy in both present-day reconstructions and future projections, and increase inter-model variability in predicted range shifts.

**Fig 2. The geographical distributions of the PNVs that are incorrect compared to the observation-based data in the model output based on the Random Forest (RF) algorithm.** Four sets of climate data were used for training and simulation: (a) averaged monthly air temperature and precipitation (*Ave*), (b) averaged monthly climate indices (*AveI*), (c) *Ave*+climate extreme indices (*CEI*), (d) *AveI*+*CEI*, (e) *Ave*+a subset of *CEI* (*CEI~part~*), and (f) *AveI*+*CEI~part~*.

Overall accuracy is a standard performance metric, but it can be misleading when class distributions are highly imbalanced, as in this study, where some biomes occupy less than 1% of grid cells. To provide a fairer assessment that accounts for chance agreement, Cohen's Kappa [33] was calculated from the confusion matrices of models trained with the *Ave* dataset. Kappa values range from 0 to 1, with 0.61–0.80 indicating "substantial" agreement and 0.81–1.00 indicating "almost perfect" agreement. The results were RF: 0.784, SVM: 0.728, and CNN: 0.759, all within the "substantial" range. The ranking by Kappa was consistent with that by test accuracy: RF > CNN > SVM.

While Cohen's Kappa offers a chance-corrected measure of overall agreement, it does not indicate the nature of classification discrepancies. To distinguish whether these discrepancies stem from differences in class proportions or from their spatial arrangement, I decomposed the errors into quantity disagreement and allocation disagreement [34]. Quantity disagreement quantifies the difference in the number of grid cells assigned to each biome class, whereas allocation

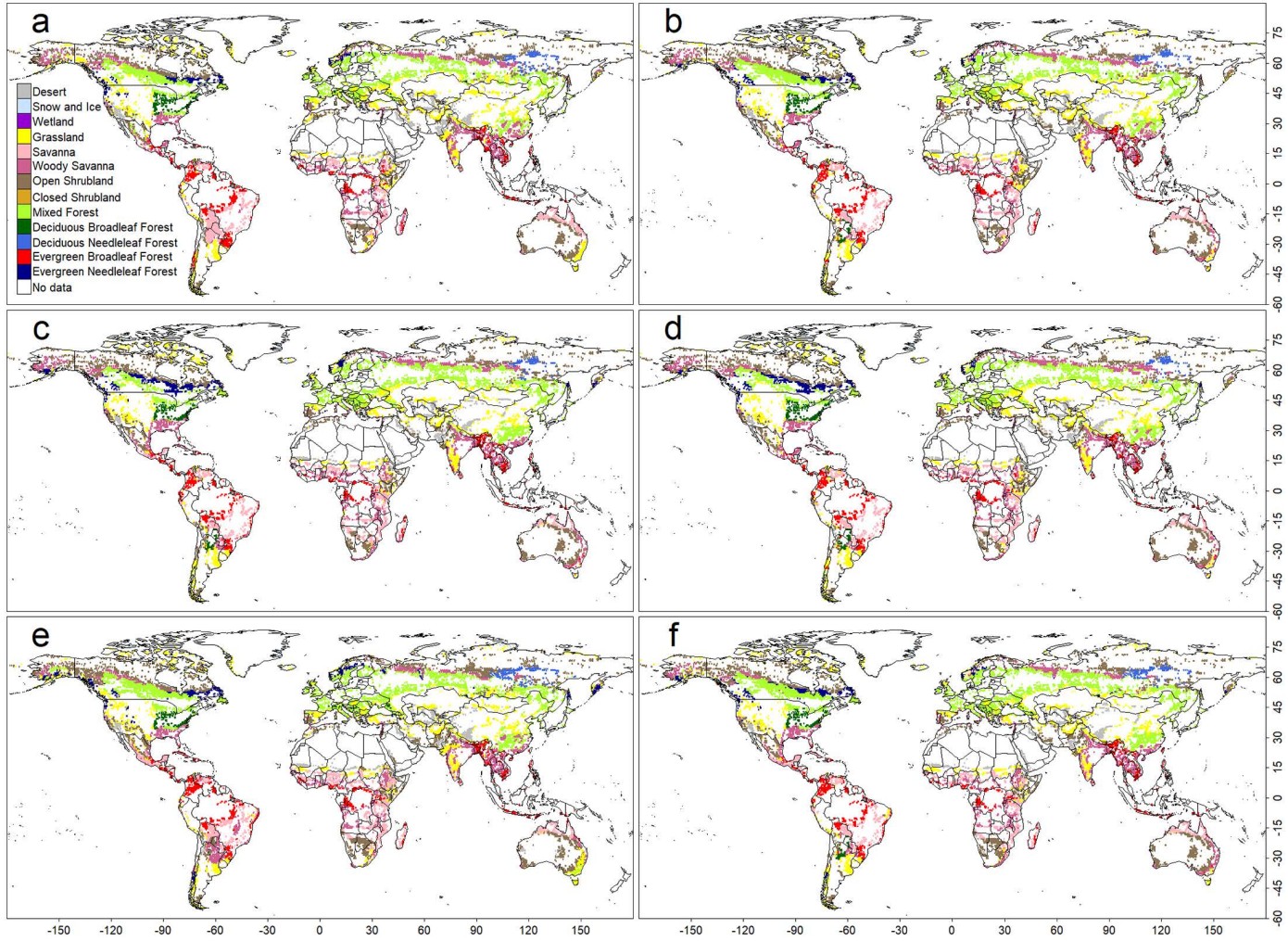

**Fig 3. As in Fig 2, but for a support vector machine (SVM)-based model.**

disagreement measures mismatches in their spatial placement. This breakdown can help guide whether model improvements should focus on adjusting overall class proportions or on refining spatial accuracy.

Quantity disagreement values were RF: 0.019, SVM: 0.045, and CNN: 0.024. Allocation disagreement values were notably higher—RF: 0.169, SVM: 0.191, and CNN: 0.185—across all models. In every case, allocation disagreement (0.169–0.191) exceeded quantity disagreement (0.019–0.045), indicating that spatial allocation errors are the primary source of model uncertainty. Consistent with other metrics, the ranking was RF > CNN > SVM. This pattern likely reflects the fragmented nature of observation-based biome distributions in regions with similar climatic conditions (Fig 1), with most mismatches occurring along biome boundaries (Figs 2, 3, and 5).

Across models, summarizing data into indices reduced test accuracy. The change from *Ave* to *AveI* was −1.1% for RF, −1.8% for SVM, and −2.0% for CNN (Table 3). Adding extreme climate indices (*CEI*) generally improved accuracy. Compared with *Ave*, *Ave+CEI* increased accuracy by +0.2% (RF), +1.6% (SVM), and +1.0% (CNN). Compared with *AveI*, *AveI+CEI* increased accuracy by +1.1% (RF), +3.1% (SVM), and +2.8% (CNN). Replacing *CEI* with partial *CEI* (*CEI_{part}*)

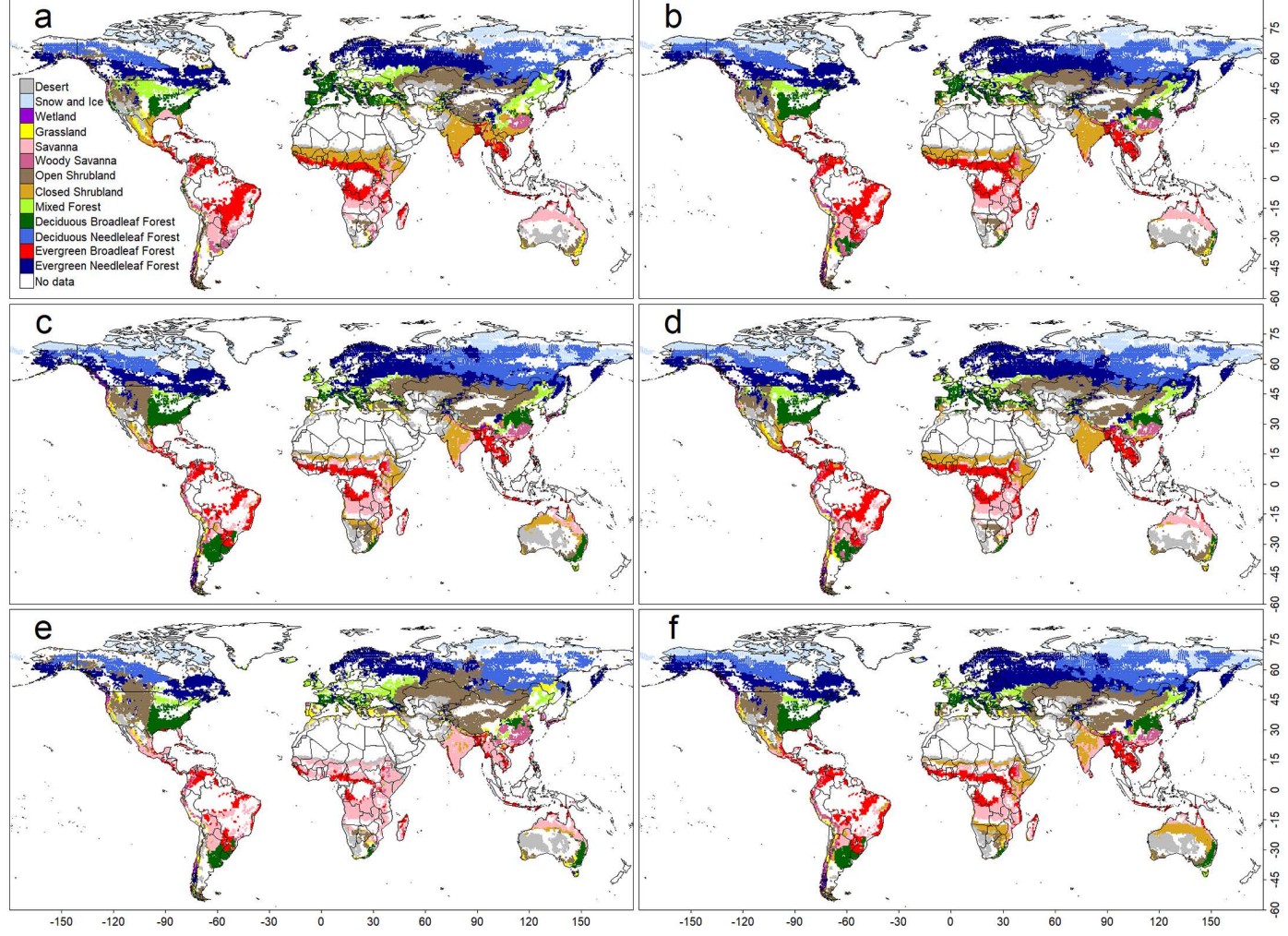

**Fig 4. As in Fig 2, but for a Naive Bayesian classifier (NV)-based model.**

produced no consistent trend: For RF, the change was negligible (+0.1% vs. + 0.1%). For SVM, accuracy decreased (−0.3% vs. −0.8%), whereas for CNN, it increased (+1.7% vs. + 2.1%).

### Training accuracy, overfitting, and future-climate consistency

For all models-datasets combinations, training accuracy exceeded test accuracy, resulting in a positive overfitting score (training accuracy − test accuracy; Table 3). The RF model consistently achieved 100% training accuracy, leading to highest overfitting scores (18.6–20.0%). SVM and CNN had much lower overfitting scores, at 1.38–2.05% and 0.75–2.17%, respectively.Under current climatic conditions, all models reconstructed highly coincident PNV distributions regardless of the training datasets (accuracy: 70.1–86.4%, Table 4). Differences in reconstructed PNV correspondence between model pairs were small: less than 2.0% between RF and SVM (accuracy: 84.5–86.4%), less than 3.3% between RF and CNN (accuracy: 70.8–74.1%), and less than 2.3% between SVM and CNN (accuracy: 70.1–72.4%).

When the trained models were applied to future climatic conditions−i.e., conditions beyond the training data−much larger differences emerged among the PNV distributions generated by different model and dataset combinations

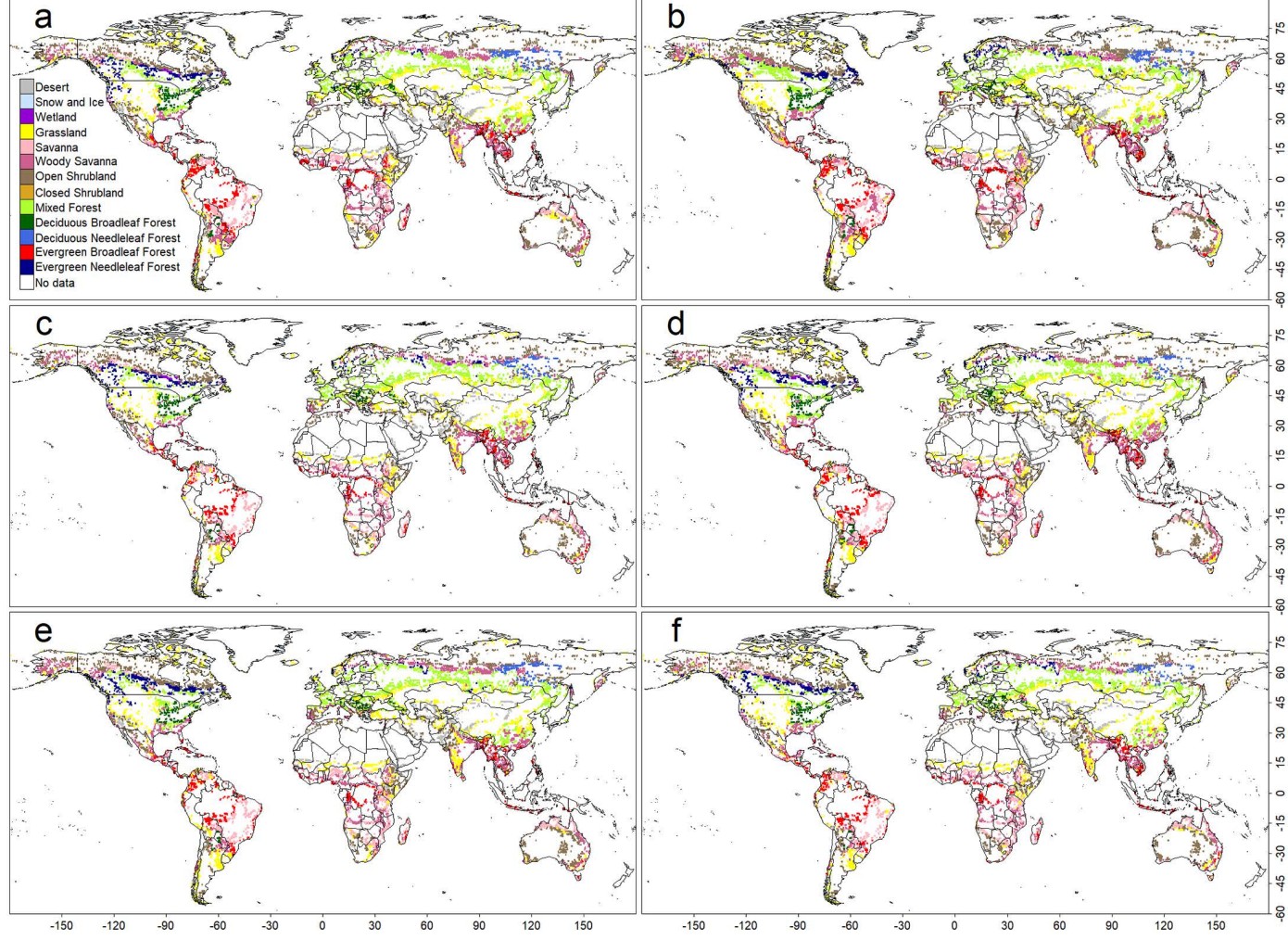

**Fig 5. As in Fig 2, but for a convolutional neural network (CNN)-based model.**

**Table 4. Degree of coincidence (%) in pairwise comparisons of simulated PNV under both current and the RCP8.5 future climate scenario. Asterisks indicate the exclusion of models with poor accuracies (i.e., excluding models trained with *CEI* as input data.). Models trained with NV were also excluded.**

|  | *Ave* | *Ave + CEI* | *Ave + CEI$_{part}$* | *Avel* | *Avel + CEI* | *Avel + CEI$_{part}$* |
|---|---|---|---|---|---|---|
| RF vs SVM | 85.6* 78.6* | 86.4 3.4 | 86.3* 82.8* | 84.4* 81.2* | 86.1 4.1 | 85.5* 82.0* |
| RF vs CNN | 70.8* 56.3* | 71.5 43.4 | 74.1* 63.1* | 70.8* 60.8* | 70.9 56.8 | 73.5* 66.3* |
| SVM vs CNN | 70.1* 65.4* | 72.3 1.6 | 70.6* 51.7* | 71.9* 65.4* | 72.4 5.1 | 70.7* 66.0* |

(accuracy: 1.6–82.8%; Table 4). Discrepancies were particularly pronounced in PNV maps from models trained on *CEI* datasets (see Figs S8–S11). SVM models trained on the *CEI* dataset predicted only evergreen broadleaf forest (Fig. SI9c, d), whereas CNN models produced maps dominated by grassland and savanna (Fig. SI11c,d). Substituting *CEI* data with partial *CEI* (*CEI$_{part}$*) reduced these extreme outputs (Figs. SI9e, f and 11e,f). When models trained with the NV algorithm and *CEI* dataset were excluded, the remaining models produced much more consistent PNV distributions under future climate conditions (accuracy: 51.7–82.8%, Table 4).

## Discussion

### Comparative performance of machine learning algorithms

Across all input dataset combinations, RF and CNN algorithms provided more accurate global PNV models than SVM and NV. A concise rationale for selecting these four algorithms and their contrasting assumptions is provided in the Methods section ("Machine learning algorithms").

Hengl, Walsh (5) found that the RF algorithm consistently outperformed other machine learning algorithms, including neural networks. In their study, a stack of 160 global maps representing biophysical conditions over the terrestrial surface—including atmospheric, climatic, relief, and lithologic variables—was used as explanatory variables to predict 20 biome classes in the BIOME 6000 dataset [35]. Although a direct comparison with the present study is not possible, their findings support RF as an effective machine learning algorithm for reconstructing biome maps. The present study is the first to compare the performance of a CNN algorithm adapted for biome modeling [6] with that of other machine learning algorithms; this CNN showed performance comparable to that of the RF algorithm.

### Model reliability: Overfitting and generalization

The RF and CNN algorithms exhibited comparable test accuracy; however, CNN was preferred because RF produced much higher overfitting scores than the other machine learning algorithms examined in this study. Although overfitting in RF could potentially be reduced by limiting tree depth, doing so would require adjusting the model based on the test data, which would compromise the experimental design by failing to keep calibration and testing independent. Overfitting is an inevitable risk associated with empirical models [32]. Fourcade, Besnard (19) demonstrated an extreme example of pseudo-predictive variables−randomly chosen classical paintings−increasing the accuracy of species distribution modeling; these models sometimes had even higher evaluation scores than models trained with relevant environmental variables. To avoid overfitting or the use of pseudo-predictive variables, Fourcade, Besnard (19) suggested investing greater effort in cross-validation and ensuring the selection of the most important predictors. This approach was followed in the present analysis.

### Model robustness under climate extremes

Adding extreme climate data slightly improved test accuracy; however, it substantially reduced model robustness, defined here as the consistency of model predictions under forecast climate conditions. This reduction in robustness was primarily driven by six *CEI* variables whose distributions deviated markedly from those in the training data, underscoring the importance of assessing the distributions of both training and prediction variables when building empirical models. Because the improvement in test accuracy obtained by including extreme climate data did not outweigh the loss in robustness, I recommend excluding extreme climate data when predicting global biome distributions at the geographical resolution used in this study (0.5°).

Here, "robustness" is operationalized as cross-algorithm consistency under future climates; while this proxy is useful for decision-making, it can also reflect reduced sensitivity to novel or extreme conditions. The observed loss of consistency was primarily driven by a small set of *CEI* variables whose distributions diverged strongly between training and projection

conditions. In practice, whether to include *CEI* should follow the research question: (i) for a high-agreement baseline map under extrapolative climates, exclude *CEI*; (ii) to explore a broader possibility space or risk envelopes, include *CEI* as a supplementary input; and (iii) for species- or local-scale questions, extremes may contribute more and could be prioritized.

### Role of climate indices in simplifying input data

The climate index data used in this study reduced the number of variables by two-thirds (from 24 to 16). However, it only slightly decreased model accuracy (−1.1%, −1.8%, and −2.0% for RF, SVM, and CNN, respectively), demonstrating that the typical climate indices employed here effectively captured extracted essential climate information relevant to global biome distribution. Nevertheless, indexing has limited utility in building machine learning-based non-transparent models, it is essential for constructing interpretable models such as decision trees [7].

### Contextualizing with previous studies and future directions

As shown above, the reliability of empirical models cannot be guaranteed beyond the range of their training data. In contrast, process-based models are expected to behave appropriately even when applied to environmental conditions that deviate slightly from those represented in observational datasets. This is one reason why many groups have proposed and developed dynamic global vegetation models (DGVMs) with greater fidelity to process to ecological processes, aiming to predict biome distributions mechanistically, considering climate, soil, and the fundamentals of plant physiology and ecology [36]. For example, Pugh, Rademacher (37) compared several DGVMs and identified discrepancies in their outputs and mechanisms, although their study did not specifically focus on biome distribution prediction. The expected increase in the frequency of extreme climate values in the future, which could significantly differ from the current distribution, may justify a shift from empirical models toward DGVMs. However, current DGVMs are not yet a reliable option for reconstructing plant population dynamic processes on a global scale; biome map predictions under commonly used climate change scenarios differ significantly among state-of-the-art DGVMs [37,38]. Therefore, empirical models continue to play an essential role in the approximate mapping of biomes under changing climatic conditions.

There is clear evidence that climate extremes influence a plant demographic processes such as growth [39,40], regeneration [41], and mortality [42,43], all of which affect plant species distributions. However, this does not necessarily mean that extreme climate data should always be included to improve biome map reconstruction, as mean climatic values are often strongly correlated with extreme climatic variables. Nevertheless, at local and species levels, extreme climate conditions may serve as more important predictors; Zimmermann, Yoccoz [44] showed that augmenting mean climate predictors with variables representing climate extremes can improves the predictive power of species distribution models.

A crucial disadvantage of the climatic envelope approach is that extrapolating current correlations between climate and biome distributions into the future may lead to substantially biased predictions. Thus, strong model performance under present climate conditions does not guarantee similar performance under novel climatic conditions that may arise in the future. However, no models—except those trained with the NV algorithm and the *CEI* dataset—showed notable increases in PNV uncertainty under projected climatic scenarios. This finding suggests that robust models can be developed beyond the training data if machine learning algorithms and climatic variables are carefully selected. The climatic envelope approach also has other limitations; for example, it ignores time lags between climate change and vegetation response, changes in atmospheric $CO_2$, and human land-use change (as discussed in Sato and Ise (6)). Nonetheless, the climatic envelope approach remains useful for various applications, including benchmarking DGVMs [36].

### Conclusion

CNN models trained on complete climate datasets without extreme indices provided the most balanced performance in terms of accuracy and robustness. This study examined how different machine learning algorithms and representations of

climate data influence the accuracy and robustness of global PNV models. Both RF and CNN algorithms produced highly accurate models; however, RF exhibited substantial overfitting, which undermined its robustness. Consequently, the CNN model was considered more reliable overall.

Summarizing climate data into indices provided minimal benefit and slightly reduced model accuracy (by 1–2% in RF, CNN, and SVM), suggesting that such simplification may be unnecessary for non-transparent models like CNN. Although incorporating extreme climate variables marginally improved accuracy (by 1–2% across the same models), it also reduced robustness—particularly under future climate conditions that deviated from those represented in the training data. These results underscore the need to weigh carefully the trade-off between model accuracy and stability when including extreme climate variables.

Overall, the findings suggest that, within the current modeling framework, CNN-based models trained on complete climate datasets without extreme indices achieve a favorable balance between predictive accuracy and generalizability in global biome modeling. Although the CNN approach uses a graphically encoded input format, the underlying climate data remains consistent across models, enabling cautious yet meaningful comparisons between algorithms.

## Supporting information

**S1 Fig. Histograms of average monthly air temperature and precipitation (*Ave*, 24 variables).** Red bars: averages for 1970–2000; Blue bars averages for 2061–2080.
(PDF)

**S2 Fig. Histograms of average monthly climate indices (*AveI*, 16 variables).** Red bars: averages for 1970–2000; Blue bars averages for 2061–2080.
(PDF)

**S3 Fig. Histograms of climate extreme indices (*CEI*, 27 variables).** Red bars: averages for 1970–2000; Blue bars averages for 2061–2080.
(PDF)

**S4 Fig. Simulated potential natural vegetation (PNV) under current climatic conditions using the RF model.** Four sets of climate data were used for training and simulation: (a) *Ave*, (b) *AveI*, (c) *Ave* + *CEI*, (d) *AveI* + *CEI*, (e) *Ave* + *CEI*$_{part}$, and (f) *AveI* + *CEI*$_{part}$.
(PDF)

**S5 Fig. Simulated PNV under current climatic conditions using the SVM model.** The same experimental setup as in S4 Fig. was used.
(PDF)

**S6 Fig. Simulated PNV under current climatic conditions using the NV model.** The same experimental setup as in S4 Fig. was used.
(PDF)

**S7 Fig. Simulated PNV under current climatic conditions using the CNN model.** The same experimental setup as in S4 Fig. was used.
(PDF)

**S8 Fig. Simulated PNV under future climatic conditions (2061–2080) projected under the IPCC RCP8.5 scenario using the RF model.** Four climate datasets were used for training and simulation: (a) *Ave*, (b) *AveI*, (c) *Ave* + *CEI*, (d) *AveI* + *CEI*, (e) *Ave* + *CEI*$_{part}$, and (f) *AveI* + *CEI*$_{part}$.
(PDF)

**S9 Fig. Simulated PNV under future climatic conditions (2061–2080) using the SVM model.** The same experimental setup as in S8 Fig. was used.
(PDF)

**S10 Fig. Simulated PNV under future climatic conditions (2061–2080) using the NV model.** The same experimental setup as in S8 Fig. was used.
(PDF)

**S11 Fig. Simulated PNV under future climatic conditions (2061–2080) using the CNN model.** The same experimental setup as in S8 Fig. was used.
(PDF)

**S1 Table. Potential Natural Vegetation (PNV) classes used in the modelings.** From the IGBP classification, three human-mediated classifications (Croplands, Cropland/Natural Vegetation Mosaics, and Urban and Built-Up Lands) and Water Bodies were neglected. Descriptions were based on Loveland and Belward.
(PDF)

**S2 Table. Confusion matrix for biome classification using the *AVE* climate data set and the RF model.** Columns represent the actual classes, while rows represent the predicted class. This matrix is based on the test grid only, with a total of 392,230 predictions from 10 independent trials (39,223 grids×10 test). Shaded diagonal cells indicate correct classifications. Each cell shows the count (top) and column-wise percentage (bottom) within the actual class.
(PDF)

**S3 Table. Confusion matric for biome classification using the SVM model.** The same dataset and evaluation as in S2 Table were used.
(PDF)

**S4 Table. Confusion matrix for biome classification using the CNN model.** The same dataset and avaluation procedure as in S2 Table were used.
(PDF)

## Acknowledgments

The author thanks the anonymous reviewers of the previous version of the manuscript. Dr. Shuntaro WATANABE (Kagoshima Univ.) and Dr. Takeshi ISE (Kyoto Univ.) offered technical support regarding issues of CNN, including the installation of the pertinent computer environments.

## Author contributions

**Conceptualization:** Hisashi Sato.

**Data curation:** Hisashi Sato.

**Formal analysis:** Hisashi Sato.

**Funding acquisition:** Hisashi Sato.

**Investigation:** Hisashi Sato.

**Methodology:** Hisashi Sato.

**Project administration:** Hisashi Sato.

**Resources:** Hisashi Sato.

**Software:** Hisashi Sato.

**Supervision:** Hisashi Sato.

**Validation:** Hisashi Sato.

**Visualization:** Hisashi Sato.

**Writing – original draft:** Hisashi Sato.

**Writing – review & editing:** Hisashi Sato.

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
