## [Decision Letter · Decision Letter 0]

27 Jun 2025

Dear Dr. Sato,

Thank you for submitting your manuscript to PLOS ONE. After careful consideration, we feel that it has merit but does not fully meet PLOS ONE’s publication criteria as it currently stands. Therefore, we invite you to submit a revised version of the manuscript that addresses the points raised during the review process.

**Please address several comments raised by the reviewers. Also, please address these issues: a). random forests classification and overfitting and hyperparameter tuning issues; b). cross validation issues; c). use of very few grid cells from the global datasets: d). providing additional accuracy metrics; e). poor predictions for some of the classes in the model; f). figure modifications, etc.**

We look forward to receiving your revised manuscript.

Kind regards,

Krishna Prasad Vadrevu, Ph.D

Academic Editor

PLOS ONE

**Journal Requirements:**

1. When submitting your revision, we need you to address these additional requirements. Please ensure that your manuscript meets PLOS ONE's style requirements, including those for file naming. The PLOS ONE style templates can be found at https://journals.plos.org/plosone/s/file?id=wjVg/PLOSOne_formatting_sample_main_body.pdf and https://journals.plos.org/plosone/s/file?id=ba62/PLOSOne_formatting_sample_title_authors_affiliations.pdf 2. We note that the grant information you provided in the ‘Funding Information’ and ‘Financial Disclosure’ sections do not match.  When you resubmit, please ensure that you provide the correct grant numbers for the awards you received for your study in the ‘Funding Information’ section. 3. Thank you for stating the following financial disclosure: Arctic Challenge for Sustainability II (ArCS II) [Program Grant Number JPMXD1420318865]   Please state what role the funders took in the study.  If the funders had no role, please state: "The funders had no role in study design, data collection and analysis, decision to publish, or preparation of the manuscript." If this statement is not correct you must amend it as needed. Please include this amended Role of Funder statement in your cover letter; we will change the online submission form on your behalf. 4. Thank you for stating the following in the Acknowledgments Section of your manuscript: This work was funded by the Arctic Challenge for Sustainability II (ArCS II) [Program Grant Number JPMXD1420318865]. The author thanks the anonymous reviewers of the previous version of the manuscript. The authors declare no conflicts of interest. We note that you have provided funding information that is not currently declared in your Funding Statement. However, funding information should not appear in the Acknowledgments section or other areas of your manuscript. We will only publish funding information present in the Funding Statement section of the online submission form. Please remove any funding-related text from the manuscript and let us know how you would like to update your Funding Statement. Currently, your Funding Statement reads as follows: Arctic Challenge for Sustainability II (ArCS II) [Program Grant Number JPMXD1420318865]  Please include your amended statements within your cover letter; we will change the online submission form on your behalf. 5. Please note that your Data Availability Statement is currently missing the repository name. If your manuscript is accepted for publication, you will be asked to provide these details on a very short timeline. We therefore suggest that you provide this information now, though we will not hold up the peer review process if you are unable. 6. We note that Figures 1 to 5 and S4 to S11 in your submission contain map images which may be copyrighted. All PLOS content is published under the Creative Commons Attribution License (CC BY 4.0), which means that the manuscript, images, and Supporting Information files will be freely available online, and any third party is permitted to access, download, copy, distribute, and use these materials in any way, even commercially, with proper attribution. For these reasons, we cannot publish previously copyrighted maps or satellite images created using proprietary data, such as Google software (Google Maps, Street View, and Earth). For more information, see our copyright guidelines: http://journals.plos.org/plosone/s/licenses-and-copyright. We require you to either present written permission from the copyright holder to publish these figures specifically under the CC BY 4.0 license, or remove the figures from your submission: a. You may seek permission from the original copyright holder of Figures 1 to 5 and S4 to S11 publish the content specifically under the CC BY 4.0 license.   We recommend that you contact the original copyright holder with the Content Permission Form (http://journals.plos.org/plosone/s/file?id=7c09/content-permission-form.pdf) and the following text:“I request permission for the open-access journal PLOS ONE to publish XXX under the Creative Commons Attribution License (CCAL) CC BY 4.0 (http://creativecommons.org/licenses/by/4.0/). Please be aware that this license allows unrestricted use and distribution, even commercially, by third parties. Please reply and provide explicit written permission to publish XXX under a CC BY license and complete the attached form.” Please upload the completed Content Permission Form or other proof of granted permissions as an "Other" file with your submission. In the figure caption of the copyrighted figure, please include the following text: “Reprinted from [ref] under a CC BY license, with permission from [name of publisher], original copyright [original copyright year].” b. If you are unable to obtain permission from the original copyright holder to publish these figures under the CC BY 4.0 license or if the copyright holder’s requirements are incompatible with the CC BY 4.0 license, please either i) remove the figure or ii) supply a replacement figure that complies with the CC BY 4.0 license. Please check copyright information on all replacement figures and update the figure caption with source information. If applicable, please specify in the figure caption text when a figure is similar but not identical to the original image and is therefore for illustrative purposes only.The following resources for replacing copyrighted map figures may be helpful: USGS National Map Viewer (public domain): http://viewer.nationalmap.gov/viewer/The Gateway to Astronaut Photography of Earth (public domain): http://eol.jsc.nasa.gov/sseop/clickmap/Maps at the CIA (public domain): https://www.cia.gov/library/publications/the-world-factbook/index.html and https://www.cia.gov/library/publications/cia-maps-publications/index.htmlNASA Earth Observatory (public domain): http://earthobservatory.nasa.gov/Landsat:
http://landsat.visibleearth.nasa.gov/USGS EROS (Earth Resources Observatory and Science (EROS) Center) (public domain): http://eros.usgs.gov/#Natural Earth (public domain): http://www.naturalearthdata.com/

Reviewers' comments:

Reviewer's Responses to Questions

**Comments to the Author**

1. Is the manuscript technically sound, and do the data support the conclusions?

Reviewer #1: Yes

Reviewer #2: No

Reviewer #3: Partly

Reviewer #4: Yes

2. Has the statistical analysis been performed appropriately and rigorously?

Reviewer #1: Yes

Reviewer #2: No

Reviewer #3: N/A

Reviewer #4: Yes

3. Have the authors made all data underlying the findings in their manuscript fully available?

Reviewer #1: Yes

Reviewer #2: Yes

Reviewer #3: Yes

Reviewer #4: Yes

4. Is the manuscript presented in an intelligible fashion and written in standard English?

Reviewer #1: Yes

Reviewer #2: Yes

Reviewer #3: No

Reviewer #4: Yes

**Reviewer #1:**  Major Revisions

The manuscript attempts to apply different machine learning techniques to classify biome under present and future climate scenarios. This research topic is relevant and holds significant potential for impactful insights. Below are some suggestions which can help improve the clarity and rigor of the manuscript.

Figures

1. For the figures 1, 2, 3, 4, and 5, please increase DPI. Use at least 300 for Image Exports. In the current form, the legend is unable to read.

2. Please include a north arrow and scale bar in the spatial maps.

Tables

1. In Table 1, please consider introducing the abbreviation for the units - C or mm - before the first appearance in the table.

2. In Table 2, the abbreviations must be redefined in the Table caption or as a footnote to provide a better readability.

3. The tables 3, 4, and 5 can be combined into a single big table to reduce the visual clutter as the three tables share a same input variable combination and 4 models. Perhaps even, using a color-coding schema for train, test and overfitting accuracies could help? This could also reduce the overall number of figures and tables.

4. The same as above can be said for Table 6 and 7

Typos

1. Page 19, Line 311: “to an RF” must be changed to “to the RF”

2. The term “pseudo-predicting” must be changed to “pseudo-predictive”. The latter seems more reasonable

Suggestions and Questions

1. While the authors acknowledge the issue of overfitting in RF, the conclusion refers to it as ‘robust’. Given that RF achieves 100 percent training accuracy and significant overfitting, calling it robust appears inconsistent with the results. It maybe helps to rephrase this statement altogether to avoid confusion.

2. I believe the authors should have done hyper parameter tuning, even if it is at minimal, for a more even model comparison.

3. Could the authors clarify if the image conversion from climatic variables performed only to train CNN or, to even test and predict? Could the authors also clarify why they chose the method from Sato and Ise? Are there any advantages to it over other methods?

4. The study lacks a feature sensitivity analysis. Thus, it is not clear which climate variables are driving predictions. If the authors note this addition, it would be helping the study.

5. The authors trained using the climate data from years 1970-2000 and then tried to project for years 2061-2080. Could the authors provide details on the intermediatory validation between the years 2000-2060 and account for this gap? This would also help to assess the model’s temporal generalization.

**Reviewer #2:**  The manuscript is relevant and shows an interesting approach. I have some suggestions for improvement:

- The first sentence of the abstract was too direct. I suggest making it more explanatory.

- In the abstract, I suggest including the accuracy value mentioned so that the phrases “marginally decreased model accuracy” and “improved accuracy slightly” are not vague. How many % of accuracy?

- The last sentence of the abstract could be left in a way that recommends not using extreme climate data. Using “should not” ends up establishing a rule, which is complicated considering only one or a few studies.

- The introduction could be more detailed and show other examples on the topic. I suggest increasing it a little and referencing studies that have already been done.

- Usually, the largest amount of data is intended for training. Why was only 25% used in this study? Using little in training could be the reason for overfitting in the end. Despite the justification and that there were tests for using 25%, have other studies adopted this methodology? Is it reproducible?

- Were there no tests performed with changes to the algorithm parameters? The use of default parameters is generally not recommended for all analyses and all algorithms. It is interesting to test whether there was any change in performance and to find the hyperparameters. It is said that default was used, but cross validation and other techniques help to ensure that the models are reliable.

- The figures and map captions are blurry. I suggest making them clearer.

- When you say that "while CNN employed graphically converted climate data, preventing a conclusive determination of the superior approach" you limit the comparison with other algorithms and it would be interesting to better justify why you continued with the 4 algorithms.

- The conclusion could be more explanatory, it is very direct.

**Reviewer #3:** Overview of the Study

This study explores the use of machine learning (ML) models to simulate and project global Potential Natural Vegetation (PNV). The models were trained on baseline climate data from 1970 to 2000 and used to predict future vegetation distributions for the period 2061–2080 under the RCP 8.5 scenario from the IPCC's Coupled Model Intercomparison Project Phase 5 (CMIP5). Four ML algorithms—Random Forest (RF), Support Vector Machine (SVM), Naïve Bayes (NB), and Convolutional Neural Network (CNN)—were tested across six configurations of four climate-related datasets, yielding 24 distinct model scenarios. These datasets included averaged monthly temperature and precipitation, averaged monthly climate indices, climate extreme indices, and a reduced version of the climate extremes dataset with six variables excluded. The author concludes that CNN achieved the best performance. The inclusion of climate indices slightly decreased model accuracy, whereas the use of climate extreme indices led to a slight improvement.

Writing and Organization

Although the manuscript has been reviewed by two native English speakers and is grammatically sound, it lacks logical structure and clarity. The flow of ideas is often difficult to follow, making it challenging for readers to grasp the author's main points. It is recommended that the manuscript be reviewed by a scientific copy editor who can improve coherence, enhance conceptual clarity, and ensure that the writing is logically organized and accessible.

Machine Learning Methodology

The methodological choices in the study raise several concerns. The global dataset consists of 52,297 grid cells, yet only 25% of these were used for training while the remaining 75% were allocated for testing. This deviates from standard ML practice, where typically 70–80% of data is reserved for training. The author repeated this process 10 times to obtain average results, but no explanation is provided for the reversed training-test split or the repetition strategy.

Additionally, the confusion matrices presented in Tables S2 to S4 lack important context. It is unclear whether they correspond to training, test, or full datasets. Examination suggests they were computed on the full dataset, as the total number of records in the matrices is 52,303, which does not match the documented dataset size of 52,297. Furthermore, the matrices are mislabeled, with both rows and columns titled "Predicted Class," without identifying the true labels, which undermines interpretability.

Evaluation Metrics and Imbalanced Data

The manuscript reports only overall accuracy as the evaluation metric across the 24 model configurations. While this metric is commonly used, it is insufficient for imbalanced datasets such as this one, which involves 13 PNV classes based on the International Geosphere-Biosphere Programme classification. Overall accuracy can obscure poor performance on minority classes and misrepresent model effectiveness. To address this issue, the author should include additional performance metrics such as class-level precision, recall, and F1 scores, as well as aggregated measures like Cohen's Kappa and metrics such as the Area Under the ROC Curve (AUC-ROC). These metrics would provide a more robust and informative comparison of model performance. The use of techniques to address class imbalance, such as SMOTE (Synthetic Minority Over-sampling Technique), is also strongly recommended.

Abstract and Supporting Materials

The abstract is not sufficiently informative. It mentions only two of the four ML algorithms and omits any reference to the comparison of the 24 model-dataset configurations. As a result, readers must progress through a significant portion of the manuscript to understand the scope of the study. The abstract should be revised to clearly summarize the methodological design and key findings.

Other Issues

Several minor but important issues were noted. The figures lack adequate resolution, and Figures 6 through 9 are referenced in line 282 but are missing from the manuscript. The citation style is inconsistent; in some cases, the first two authors are named followed by a reference number, even when more than three authors are involved. Long URLs and access dates appear in the body of the text and should instead be properly formatted as references, in accordance with journal guidelines. There is also an inconsistency in the definition of the future projection period: while the study defines this period as 2061–2080, Supplementary Figures S8 to S11 appear to show data for the year 2100. This discrepancy requires clarification. Finally, the author does not justify the choice of CMIP5 data over the more recent CMIP6 datasets, which are increasingly favored in contemporary climate studies. Providing a rationale for this choice is essential.

**Reviewer #4:** The author presents a study on modelling biome distribution at global scale using machine learning. The performances of four machine-learning algorithms driven by six combinations of four current climate datasets are compared. The performances of each of the 24 resulting models (i.e., algorithm-climate dataset combinations) are assessed based on its ability to reproduce the current observation-based potential vegetation map produced by Beigaite et al. (2022) with limited overfitting, and its consistency with other models when predicting future potential natural vegetation under a Representative Concentration Pathway 8.5 (RCP8.5) climate scenario. Based on such comparisons of model performances, the author recommends using the CNN algorithm from Sato and Ise (2022) and excluding extreme climate data when building models to predict global-scale biome distribution.

I believe this is an interesting and timely study that aims to clarify and compare the current potential of different machine learning methods for predicting vegetation distribution on a global scale. Overall, while I think the study has the potential to be published, several major revisions are required. I find that the manuscript would particularly benefit from (1) a clarification of the research question, (2) a better contextualisation, and (3) a better description and justification for the methods, products, and programs used. I also find that the manuscript tends to rely too little on the literature, which leads to a certain amount of unsourced methodological choices and statements.

Major comments

(1) Stating a clear question along with some hypotheses from start would improve the readability. Several questions are addressed, and I'm having trouble identifying the real focus: is the priority to identify the best-performing ML algorithm, to assess the value of not summarising climate data into climate indices, to study the potential of incorporating extreme climate indices, or the potential of machine learning to predict the future distribution of biomes in the context of climate change?

(2) To better emphasise the value of the issues addressed in this study, I would find it helpful to describe from the start the general approach of machine learning and the interest of using it to predict the distribution of vegetation. In particular, what are the advantages of using machine learning compared with other existing approaches such as other correlative models or more process-based models. For example, Beigaite et al. (2022) and Bonannella et al. (2023) provide some details in their introductions that could be mentioned here. It is suggested on L40-52 that the main advantage of machine learning is the ability to use a larger amount of climate data, but it is not stated how this gives the machine learning an advantage in terms of its performance over other approaches. In general, I find it difficult to identify use cases for such models, given that biome/land-cover maps derived from satellite observations seem more relevant for mapping current distribution, and that the present manuscript demonstrates the difficulties of such models for predicting future vegetation distribution on a global scale. This point requires further attention and justification.

Bonannella, C., T. Hengl, L. Parente, and S. de Bruin. 2023. Biomes of the world under climate change scenarios: increasing aridity and higher temperatures lead to significant shifts in natural vegetation. PeerJ 11:e15593.

(3) Most of the products and programs used are treated as “black boxes” (machine learning algorithms and their default parameters, the potential natural vegetation map, climate data). Generally, little or nothing is said about the data and methods used to construct and validate them, the assumptions associated with them and the reasons for selecting them for this study. I find that this blurs the implications and conclusions of this study.

(4) The manuscript would benefit from a better description of the fundamental similarities and differences between the four machine learning approaches compared in this study, and the hypotheses associated with them. Hengl et al. (2018) included other machine learning algorithms in the comparison (from Friedman 2002, and Venables and Ripley 2002). Why choose to compare these four in particular? So far, although some specific aspects are being discussed, the algorithms generally appear to be black boxes and it is difficult to understand the origin of the differences observed. Before drawing general conclusions, it would be interesting to detail, at least in the form of hypotheses, the potential reasons why a model outperforms another, and what does it mean for the associated sets of data and assumptions.

Friedman, J. H. 2002. Stochastic gradient boosting. Computational Statistics & Data Analysis 38:367–378.

Venables, W. N., and B. D. Ripley. 2002. Modern applied statistics with S (Fourth Edition). New York: Springer-Verlag.

(5) The performances of the models are compared against their ability to predict the PNV map of Beigaite et al. (2022) which is itself derived from IGBP-MODIS from Friedl et al. (2010). The reasons for this choice are not given, although there are many other biome maps and classifications with numerous differences (reviewed for example in Mucina 2019, Beierkuhnlein and Fischer 2021, Hunter et al. 2021, Fischer et al. 2022, Champreux et al. 2024). After reading the manuscript, the potential effects of the choice of this product on the results and conclusions remain unclear.

Beierkuhnlein, C., and J.-C. Fischer. 2021. Global biomes and ecozones – Conceptual and spatial communalities and discrepancies. Erdkunde 75:249–270.

Champreux, A., F. Saltré, W. Traylor, T. Hickler, and C. J. A. Bradshaw. 2024. How to map biomes: Quantitative comparison and review of biome-mapping methods. Ecological Monographs 94:e1615.

Fischer, J.-C., A. Walentowitz, and C. Beierkuhnlein. 2022. The biome inventory – Standardizing global biogeographical land units. Global Ecology and Biogeography 31:2172–2183.

Hunter, J., S. Franklin, S. Luxton, and J. Loidi. 2021. Terrestrial biomes: a conceptual review. Vegetation Classification and Survey 2:73–85.

Mucina, L. 2019. Biome: evolution of a crucial ecological and biogeographical concept. New Phytologist 222:97–114.

(6) The unequal spatial coverage of the different biomes does not appear to be considered in the manuscript. The model performances are assessed via the percentage of correct answers. Because of its simplicity, such a measure seems to facilitate interpretation. However, it does not consider the variability of the areas covered by the different biomes. The choice of this metric over other common map comparison metrics such as Cohen’s kappa (Cohen 1960, Monserud and Leemans 1992) or the quantity-and-allocation agreement (Pontius and Millones 2011) thus requires better justification. It also seems obvious to me that the models should tend to better predict the most extensive biomes on a global scale since the models necessarily receive more data on them during the training phase. This is perhaps one of the potential reasons for the poor performance of the models in predicting the “Wetland” and “Closed Shrubland” biomes as stated on L236-244, respectively representing 0.5 and 0.2 % of the grid cells according to S1 table. The differences observed in future predictions are also likely to be affected by this unequal distribution of biomes. While the poor prediction of biomes covering smaller total areas globally affects current predictions less, their potential expansion in the future is likely to be poorly captured and to mechanically increase the disparities among models.

Cohen, J. 1960. A Coefficient of Agreement for Nominal Scales. Educational and Psychological Measurement 20:37–46.

Monserud, R. A., and R. Leemans. 1992. Comparing global vegetation maps with the Kappa statistic. Ecological Modelling 62:275–293.

Pontius, R. G., Jr, and M. Millones. 2011. Death to Kappa: birth of quantity disagreement and allocation disagreement for accuracy assessment. International Journal of Remote Sensing 32:4407–4429.

Minor comments

(7) L18. I would find it helpful to mention the four machine learning algorithms compared in the abstract.

(8) L40. “To date, many methods have been proposed to model biome distributions” Sato and Ise (2022) is cited here to support this statement, but also citing other sources reviewing the various methods to model biome distributions in greater details would be more appropriate. Generally, the statements in this paragraph should refer more to existing literature.

(9) L84-85. I wonder why eliminating only grid cells with 100 % human activity. Keeping in the analysis other highly impacted areas could possibly strongly affect the outputs. For example, Champreux et al. (2024) suggested that human activities could cause disagreement among biome maps including among PNV maps. Anyway, providing the percentage of grid cells that were eliminated due to the “100 % human activity” criteria would be helpful.

(10) L130-140. The models are trained with current, observation-based climate datasets but future predictions are made with simulated data that are associated with more uncertainties. Although these methodological differences are obviously unavoidable, I wonder whether they are likely to bias the assessment of model performances in future predictions. For example, it is not stated if the variables used have a standard definition that is common to both current and future datasets, or if some equivalencies have been performed.

(11) L134-135. “The Representative Concentration Pathway (RCP) 8.5 was the only used.” The choice of this single scenario should be justified.

(12) L144. Reference for CNN (Sato and Ise, 2022) is missing.

(13) L150-151. The author state “Simplicity was maintained and potential overfitting was mitigated by using the default settings in these commands”. This statement needs to be better justified.

(14) L154-158. This statement needs to be associated with supporting literature.

(15) L181. While I understand that model robustness was assessed based on the consistency among models for future predictions, this part of the analysis is not described in the method section, except for the presentation of the BIOCLIM data on L130-140. Describing it in the “Data analysis” section would be appropriate.

(16) L190. It is unclear if the selected value of 25% corresponds to standards of the discipline or was decided due to other reasons. Justifying it is important as it probably impacts the differences in overfitting scores of the models in the end, especially considering the 100% training accuracy of the RF models mentioned on L256-257.

(17) L199-200. “precisely” should be defined, as no decision threshold was provided in the method section.

(18) L204-205. “all grids at the same latitude were assigned the most frequent PNV at that latitude” This statement is unclear.

(19) L242. I guess that “The incorrectly identified biomes” is a typo that should be replaced with “The incorrectly identified grid cells”.

(20) L298-301. “Default parameter settings were used for all methods adopted in this study, and the models except the CNN were trained with climate data, while CNN employed graphically converted climate data, preventing a conclusive determination of the superior approach.” The implications of this methodological difference are unclear. It would be interesting to elaborate a bit more on this.

(21) Did Hengl et al. (2018) consider this overfitting issue with the RF algorithm?

(22) L322-325. If this statement means that the RF model is still showing strong overfitting despite these considerations, this calls into question the recommendations of Fourcade et al. (2018) and deserves to be discussed.

(23) L335-338. The consistency among future predictions is considered here as a proxy of model robustness. However, it also highlights the weaknesses of the machine learning algorithms for predicting future biome distributions where extremes are supposed to become more frequent and unprecedented. This point is discussed in the subsequent paragraphs, but the use of the consistency metric to assess model robustness is not questioned.

(24) L351-353. This is a rather strong statement justifying the use of machine learning over DGVMs for future biome map predictions. The statement is only supported here by citing Pugh et al. (2020). However, while Pugh et al. (2020) did compare several DGVMs and show several output and mechanism discrepancies, the study did not focus on predicting biome distribution, and did not provide such a conclusion.

(25) L364-367. This statement may be helpful in the introduction section to justify testing the incorporation of extreme climate variables for predictions at larger spatial scales (here biomes).

(26) L383-384. “thus, the CNN model is preferable”. This conclusion seems paradoxical given the earlier claim suggesting that CNN's performance is difficult to compare with that of other models, on L299-301: “the models except the CNN were trained with climate data, while CNN employed graphically converted climate data, preventing a conclusive determination of the superior approach.”

**Do you want your identity to be public for this peer review?** For information about this choice, including consent withdrawal, please see our Privacy Policy

Reviewer #1: No

Reviewer #2: No

Reviewer #3: No

Reviewer #4: No

---

## [Author Response · Author response to Decision Letter 1]

13 Aug 2025

Dear Editor and Reviewers,

I sincerely thank the reviewers for their constructive comments, and I greatly appreciate your support and patience throughout the review process. I have revised the manuscript in accordance with the reviewers' suggestions and comments. Please find below my point-by-point responses addressing each concern raised.

In this letter, the line numbers in my responses refer to the clean version of the revised manuscript (i.e., the version without track changes). Throughout this letter, quoted comments from the reviewers are indicated by lines beginning with ">"”.

Kind regards,

Hisashi Sato (Author)

Revisions according to the editor's comments

1. Manuscript style

I have revised manuscript so that it meets PLOS ONE's style requirements, including those for file naming.

2. Funding

I removed funding-related text from the manuscript. Please use following statement for the grant information:

This work has been financed by the Arctic Challenge for Sustainability II (ArCS II) [Program Grant Number JPMXD1420318865]. I received no additional external funding for this study. The funders had no role in study design, data collection and analysis, decision to publish, or preparation of the manuscript.

3. Data Availability Statement

Sorry for not stating the repository name in the Data Availability Statement. Please use the following phrase for this section:

All dataset files are available from the online repository: https://doi.org/10.5281/zenodo.8113935 (Zenodo data repository).

4. Copyright of map images in figures

I created these world maps using the R library "maps". The world map data included in the maps package is derived from public domain sources, and there are no copyright restrictions on general use or redistribution. The package's online documentation (https://cran.r-project.org/web/packages/maps/maps.pdf) includes the following statement on page 32, under "Description": This world map (updated in 2013) is imported from the public domain Natural Earth project (the 1:50m resolution version).

One-to-one response to comments by Reviewer #1

> The manuscript attempts to apply different machine learning techniques to classify biome under present and future climate scenarios. This research topic is relevant and holds significant potential for impactful insights. Below are some suggestions which can help improve the clarity and rigor of the manuscript.

Thank you for your effort in reviewing my manuscript.

I carefully studied your comments and addressed them as follows.

> Figures

> 1. For the figures 1, 2, 3, 4, and 5, please increase DPI. Use at least 300 for Image Exports. In the current form, the legend is unable to read.

I appreciate the reviewer's comment regarding the low resolution of Figures in the submitted PDF. The reduced quality appears to have resulted from the journal’s submission system automatically converting the manuscript and figures into a single PDF, which down-sampled the images during processing. The original image files meet or exceed 300 dpi. In the revised submission, I have re-uploaded all figures as separate high-resolution files (>=300 dpi) to ensure that the final published version maintains full clarity and readability.

> 2. Please include a north arrow and scale bar in the spatial maps.

Since the maps provided are global-scale Mercator projections with clearly indicated latitude and longitude lines, the direction (north upward) is universally implicit, and the scale varies significantly with latitude, making a single scale bar misleading. Therefore, I opted not to include a north arrow or scale bar, following standard cartographic practice.

> Tables

> 1. In Table 1, please consider introducing the abbreviation for the units - C or mm - before the first appearance in the table.

I have added a brief note below Table 1 to clarify units used.

Note: Units: °C = degrees Celsius; mm = millimeters.

> 2. In Table 2, the abbreviations must be redefined in the Table caption or as a footnote to provide a better readability.

I want to clarify that Table 2 does not contain any abbreviations that require definition. The letter combinations (e.g., FD) used in this table represent variable IDs, assigned solely for identification purposes, rather than abbreviations. Although the IDs seem to originate from descriptive terms (e.g., 'FD' from 'Frost Days'), they are intended only as identifiers in this context, and thus, explicit definitions of their origin in the caption or footnotes are unnecessary. I want your understanding regarding this decision.

> 3. The tables 3, 4, and 5 can be combined into a single big table to reduce the visual clutter as the three tables share a same input variable combination and 4 models. Perhaps even, using a color-coding schema for train, test and overfitting accuracies could help? This could also reduce the overall number of figures and tables.

I merged Tables 3-5 into a single table as recommended, revised the caption accordingly, and updated the corresponding in-text references. (Table 3 )

> 4. The same as above can be said for Table 6 and 7

I followed the same approach for Tables 6 and 7: they were combined into a single table, the caption was revised, and all in-text references were updated accordingly. (Table 4)

> Typos

> 1. Page 19, Line 311: "to an RF" must be changed to "to the RF"

Addressed (L378).

> 2. The term "pseudo-predicting" must be changed to "pseudo-predictive". The latter seems more reasonable

Addressed (L387, 390).

> Suggestions and Questions

> 1. While the authors acknowledge the issue of overfitting in RF, the conclusion refers to it as 'robust'. Given that RF achieves 100 percent training accuracy and significant overfitting, calling it robust appears inconsistent with the results. It maybe helps to rephrase this statement altogether to avoid confusion.

I agree with the inconsistency regarding the use of the term "robust" in describing the RF model performance. I have replaced the term "robust" with "effective," acknowledging the presence of overfitting while more accurately highlighting the model's predictive capabilities.

Previous (L306-308):

Although a direct comparison with the findings of the current study is impossible, this previous report supports RF as a robust machine learning algorithm for reconstructing biome maps.

Revised (L373-375):

Although a direct comparison with the present study is not possible, their findings support RF as an effective machine learning algorithm for reconstructing biome maps.

> 2. I believe the authors should have done hyper parameter tuning, even if it is at minimal, for a more even model comparison.

In the Machine learning algorithms subsection, I added sentences to clarify this rationale.

Inserted (L105-112):

All models were run with default settings to (1) ensure fair comparability by avoiding bias from parameter tuning, (2) keep the implementation straightforward and reproducible, and (3) align with the study’s objective of evaluating algorithms and data combinations rather than optimizing a single best-performing model. This choice also simplified implementation. The RF model, for instance, already achieved 100% accuracies on the training sets with default parameters, indicating strong overfitting and suggesting that further tuning would not improve generalization in this case.

> 3. Could the authors clarify if the image conversion from climatic variables performed only to train CNN or, to even test and predict? Could the authors also clarify why they chose the method from Sato and Ise? Are there any advantages to it over other methods?

I clarified in the revised manuscript that the image conversion was applied not only during training but also during testing and prediction to ensure consistency in the CNN modeling process. I also elaborated on the rationale for choosing the method proposed by Sato and Ise (2022).

Previous (L163-164):

This method represents climatic conditions using graphical images and employs them as training data for CNN models.

Revised (L124-130):

This method represents climatic conditions using graphical images and employs them as training, testing, and prediction data for CNN models. I selected this method because it allows CNNs-originally developed for image analysis-to automatically extract nonlinear seasonal patterns from multiple climate variables while preserving their temporal structure, enabling convolutional filters to identify spatially coherent features. This method automatically extracts nonlinear seasonal patterns for climatic variables relevant to biome classification.

> 4. The study lacks a feature sensitivity analysis. Thus, it is not clear which climate variables are driving predictions. If the authors note this addition, it would be helping the study.

I agree that feature sensitivity analysis is valuable in many contexts. However, the primary focus of this study was to evaluate (1) differences in performance among machine learning algorithms, (2) the effect of summarizing climate data into indices, and (3) the impact of incorporating extreme climate indices. Identifying the importance of specific climate variables was beyond the scope of this research.

Furthermore, as the CNN model used in this study relies on automatically extracted features from image-based inputs, conventional feature importance analysis is not readily applicable.

> 5. The authors trained using the climate data from years 1970-2000 and then tried to project for years 2061-2080. Could the authors provide details on the intermediatory validation between the years 2000-2060 and account for this gap? This would also help to assess the model’s temporal generalization.

I agree that intermediate validation could be informative in other contexts. However, as noted in the revised manuscript, this study was designed to test model robustness under maximum climate divergence. Therefore, only the RCP8.5 scenario and its end-of-century projection were used to fulfill this specific goal.

Inserted (L200-203):

Intermediate projection periods (e.g., 2041-2060) were not used because the aim was to test model robustness under the most extreme climate conditions. Since RCP8.5 represents the highest emission trajectory, its far-future projection.

One-to-one response to comments by Reviewer #2

> The manuscript is relevant and shows an interesting approach. I have some suggestions for improvement:

Thank you for your effort in reviewing my manuscript.

I carefully studied your comments and addressed them as follows.

> The first sentence of the abstract was too direct. I suggest making it more explanatory.

Indeed, the original sentence may have been too concise and direct, making the background and context of the study unclear to readers. Therefore, we revised the opening to first address the importance of the research topic before referring to previous studies.

Previous (L12-13):

Many methodologies have been proposed for modeling global biome distributions.

Revised (L12-13):

Understanding the global distribution of biomes is essential for biodiversity conservation, climate modeling, and land-use planning.

> In the abstract, I suggest including the accuracy value mentioned so that the phrases "marginally decreased model accuracy" and "improved accuracy slightly" are not vague. How many % of accuracy?

I revised the abstract and the conclusion to include specific accuracy changes (-1~2% for summarization, +1~2% for extreme indices), based on the RF, CNN, and SVM models.

Previous (L25-26):

Summarization of climate data into indices marginally decreased model accuracy, whereas incorporating extreme climate indices improved accuracy slightly.

Revised (L20-22):

Summarizing climate data into indices reduced accuracy by 1-2%, while adding extreme indices increased accuracy by <2% (except for NV, which performed poorly overall).

> The last sentence of the abstract could be left in a way that recommends not using extreme climate data. Using "should not" ends up establishing a rule, which is complicated considering only one or a few studies.

To shift from a prescriptive to a more suggestive tone and to allow room for reader interpretation, I revised the sentence as follows.

Previous (L29-31):

Based on these findings, extreme climate data should not be included in global-scale biome prediction models due to their detrimental impact on model robustness.

Revised (L24-26):

These results indicate that including extreme climate data in global biome prediction models offers limited accuracy gains but can significantly weaken robustness, so caution is advised.

> The introduction could be more detailed and show other examples on the topic. I suggest increasing it a little and referencing studies that have already been done.

I added a new paragraph (L57-74) in the Introduction for clarifying the advantages of machine learning over process-based models and land cover maps. I believe this addition sufficiently addresses the reviewer's request for more context and examples of previous work.

> Usually, the largest amount of data is intended for training. Why was only 25% used in this study? Using little in training could be the reason for overfitting in the end. Despite the justification and that there were tests for using 25%, have other studies adopted this methodology? Is it reproducible?

I added a sentence in the Methods section to clarify why a 25% training ratio was used in this study.

Inserted (L215-218)

This proportion is the reverse of the typical 70-80% allocation to training [29], chosen to emphasize model robustness over performance and to ensure that rare vegetation types (<1%) were adequately represented in the test set.

> Were there no tests performed with changes to the algorithm parameters? The use of default parameters is generally not recommended for all analyses and all algorithms. It is interesting to test whether there was any change in performance and to find the hyperparameters. It is said that default was used, but cross validation and other techniques help to ensure that the models are reliable.

In the Machine learning algorithms subsection, I added a sentences to clarify this rationale.

Inserted (L105-114)

All models were run with default settings to (1) ensure fair comparability by avoiding bias from parameter tuning, (2) keep the implementation straightforward and reproducible, and (3) align with the study’s objective of evaluating algorithms and data combinations rather than optimizing a single best-performing model. This choice also simplified implementation. The RF model, for instance, already achieved 100% accuracies on the training sets with default parameters, indicating strong overfitting and suggesting that further tuning would not improve generalization in this case. While strategies such as cross-validation and variable selection can reduce overfitting [19], the RF results indicate that high-capacity models may still overfit even under best-practices settings.

> The figures and map captions are blurry. I suggest making them clearer.

I appreciate the reviewer's comment regarding the low resolution of Figures in the submitted PDF. The reduced quality appears to have resulted from the journal’s submission system automatically converting the manuscript and figures into a single PDF, which down-sampled the images during processing. The original image files meet or exceed 300 dpi. In the revised submission, I have re-uploaded all figures as separate high-resolution files (>=300 dpi) to ensure that the final published version maintains full clarity and readability.

> When you say that "while CNN employed graphically converted climate data, preventing a conclusive determination of the superior approach" you limit the comparison with other algorithms and it would be interesting to better justify why you continued with the 4 algorithms.

I deleted the sentence suggesting limited comparability of CNN, as earlier text already clarifies that the same underlying information was used. I also added brief notes to explain the rationale for

---

## [Decision Letter · Decision Letter 1]

17 Nov 2025

Dear Dr. Sato,

Thank you for submitting your manuscript to PLOS ONE. After careful consideration, we feel that it has merit but does not fully meet PLOS ONE’s publication criteria as it currently stands. Therefore, we invite you to submit a revised version of the manuscript that addresses the points raised during the review process.

We look forward to receiving your revised manuscript.

Kind regards,

Chong Xu

Academic Editor

PLOS ONE

Journal Requirements:

Reviewers' comments:

Reviewer's Responses to Questions

**Comments to the Author**

Reviewer #1: All comments have been addressed

Reviewer #4: All comments have been addressed

2. Is the manuscript technically sound, and do the data support the conclusions?

Reviewer #1: Yes

Reviewer #4: Yes

3. Has the statistical analysis been performed appropriately and rigorously?

Reviewer #1: Yes

Reviewer #4: Yes

4. Have the authors made all data underlying the findings in their manuscript fully available?

Reviewer #1: Yes

Reviewer #4: Yes

5. Is the manuscript presented in an intelligible fashion and written in standard English?

Reviewer #1: Yes

Reviewer #4: Yes

Reviewer #1: I thank the author for addressing my previous comments, and I am satisfied with the responses. The figures and tables have been improved, and the manuscript is now clearer and sound. I recommend acceptance.

Reviewer #4: The author presents a study on modelling biome distribution at global scale using machine learning under present and future climate scenarios. This is the second time I read the manuscript, which has been largely revised, greatly improving its quality compared to the previous version. The author has made an effort to respond carefully to all comments. In particular, the overall research questions are stated more clearly and most of the methodological choices are now better justified.

I only have a few minor comments :

- L86-88. I still question the relevance of retaining in the analysis grid cells that are highly impacted by human activities but do not reach 100%. The manuscript states that "grid cells partially affected by human activity were retained, on the assumption that the relative proportions of natural vegetation remain stable despite these changes. However, it has been shown that PNV maps disagree when human activities are high, suggesting that the proportion of remaining natural vegetation is not sufficiently informative in these grid cells. I therefore find the justification insufficient.

- L93-94. While the manuscript explains the importance of using a PNV map over maps derived from satellite imagery, and describes the product used, the choice of this specific PNV map for this study over others is insufficiently justified. The statement « this dataset was selected because it is particularly well suited for global climate-vegetation modeling » is not associated to any argument.

- L352-366. The paragraph justifying the choice of these four specific ML algorithms and explaining their differences appears in the Discussion section. As it highlights one of the strengths of the study and illustrates well how the manuscript adresses the overall research questions, it would be more appropriate in the ‘Methods’ section under ‘Machine learning algorithms’.

- L392-402. The manuscript recommends not including extreme climate data since it reduces robustness (here, consistency among models) under forecast climate conditions. However, the differences observed in predictions of the future state of vegetation also make it possible to define a universe of possibilities and therefore seem rather interesting. The fact that this extreme climate data improves test accuracy, even slightly, makes this observation even more interesting. The possibility that only one of these models accurately predicts the response of biome distribution to climate change remains plausible, although unprovable. The choice of whether to include or exclude this data therefore seems to me to depend on the research question, and would thus benefit from more discussion.

**Do you want your identity to be public for this peer review?** For information about this choice, including consent withdrawal, please see our Privacy Policy

Reviewer #1: No

Reviewer #4: No

---

## [Author Response · Author response to Decision Letter 2]

3 Dec 2025

Reviewer #1:

> I thank the author for addressing my previous comments, and I am satisfied with the responses. The figures and tables have been improved, and the manuscript is now clearer and sound. I recommend acceptance.

Response:

Thank you very much for your positive evaluation and supportive comments.

I sincerely appreciate your careful reading of the revised manuscript and your recognition of the improvements.

Reviewer #4:

> The author presents a study on modelling biome distribution at global scale using machine learning under present and future climate scenarios. This is the second time I read the manuscript, which has been largely revised, greatly improving its quality compared to the previous version. The author has made an effort to respond carefully to all comments. In particular, the overall research questions are stated more clearly and most of the methodological choices are now better justified.

Response:

Thank you very much for your positive evaluation and supportive comments.

I sincerely appreciate your careful reading of the revised manuscript and your recognition of the improvements.

Below, I provide point-by-point responses to each of your comments.

(1) Comment on L86-88.

> I still question the relevance of retaining in the analysis grid cells that are highly impacted by human activities but do not reach 100%. The manuscript states that "grid cells partially affected by human activity were retained, on the assumption that the relative proportions of natural vegetation remain stable despite these changes. However, it has been shown that PNV maps disagree when human activities are high, suggesting that the proportion of remaining natural vegetation is not sufficiently informative in these grid cells. I therefore find the justification insufficient.

Response:

In Methods (section "Biome data"), I added a brief explanation of the study aim-estimating PNV at 0.5°-and why grid cells with partial human influence still retain an informative dominant natural-vegetation signal at this scale. I also specified the exclusion of cells with 100% human cover and/or water and the corresponding exclusion ratio, and explicitly noted increased uncertainty in heavily impacted regions.

[Inserted text (L102-109)]

Because the aim of this study is to model potential natural vegetation (PNV), I retained grid cells that are only partially affected by human activity, on the premise that the dominant natural-vegetation signal remains informative at 0.5° resolution. By contrast, cells with 100% human cover and/or water were excluded, leaving 52,297 land grid cells (approximately 10-11% of land cells excluded, Antarctica removed). This choice minimizes spatial coverage bias while preserving information on the prevailing natural type. I acknowledge, however, that uncertainty in PNV estimation may increase in regions with strong human influence, and I interpret results in those areas with caution.

(2) Comment on L93-94.

> While the manuscript explains the importance of using a PNV map over maps derived from satellite imagery, and describes the product used, the choice of this specific PNV map for this study over others is insufficiently justified. The statement <<this dataset was selected because it is particularly well suited for global climate-vegetation modeling >> is not associated to any argument.

Response:

In Methods, I added sentences giving practical reasons for selecting this PNV dataset (alignment with our analysis resolution, consistent processing with AveI/CEI and future projections, and improved comparability with related studies).

[Inserted text (L94-101)]

I used this PNV dataset for three practical reasons aligned with global climate-vegetation modelling at 0.5°: (i) the workflow-deriving the dominant natural class from MODIS/IGBP and resampling to ~50-km grids-matches my analysis grid; (ii) its climate inputs and projections are processed consistently with the BIOCLIM (AveI) and CLIMDEX (CEI) indices used here, with harmonized definitions and resolution for present data and CMIP5-based futures; and (iii) employing the same dataset as related machine-learning studies facilitates comparability without additional preprocessing.

(3) Comment on L352-366.

> The paragraph justifying the choice of these four specific ML algorithms and explaining their differences appears in the Discussion section. As it highlights one of the strengths of the study and illustrates well how the manuscript adresses the overall research questions, it would be more appropriate in the ‘Methods’ section under ‘Machine learning algorithms’.

Response:

As suggested, I kept only the opening performance-summary sentence in the Discussion and moved the subsequent rationale and contrasts among the algorithms to Methods (Machine learning algorithms). I also added a one-sentence pointer in the Discussion directing readers to Methods; the substantive content itself was unchanged.

[Inserted text (L382-384)]

A concise rationale for selecting these four algorithms and their contrasting assumptions is provided in the Methods section ("Machine learning algorithms").

(4) Comment on L392-402.

> The manuscript recommends not including extreme climate data since it reduces robustness (here, consistency among models) under forecast climate conditions. However, the differences observed in predictions of the future state of vegetation also make it possible to define a universe of possibilities and therefore seem rather interesting. The fact that this extreme climate data improves test accuracy, even slightly, makes this observation even more interesting. The possibility that only one of these models accurately predicts the response of biome distribution to climate change remains plausible, although unprovable. The choice of whether to include or exclude this data therefore seems to me to depend on the research question, and would thus benefit from more discussion.

Response:

To avoid redundancy, I deleted the original closing sentence of the paragraph and replaced it with three sentences that (i) define our operational use of robustness, (ii) attribute the loss of consistency to CEI variables with strong distributional shifts, and (iii) provide purpose-dependent guidance on when to include or exclude CEI.

[Deleted L400-402 of the cleaned previous manuscript]

Although consistency is used here as a proxy for robustness, it may also indicate a lack of sensitivity to novel or extreme conditions.

[Inserted text (L419-427)]

Here, "robustness" is operationalized as cross-algorithm consistency under future climates; while this proxy is useful for decision-making, it can also reflect reduced sensitivity to novel or extreme conditions. The observed loss of consistency was primarily driven by a small set of CEI variables whose distributions diverged strongly between training and projection conditions. In practice, whether to include CEI should follow the research question: (i) for a high-agreement baseline map under extrapolative climates, exclude CEI; (ii) to explore a broader possibility space or risk envelopes, include CEI as a supplementary input; and (iii) for species- or local-scale questions, extremes may contribute more and could be prioritized.

---

## [Editor Report · Decision Letter 2]

10 Dec 2025

Predicting dominant terrestrial biomes at a global scale using machine learning algorithms, climate variable indices, and extreme event indices

PONE-D-25-21376R2

Dear Dr. Sato,

We’re pleased to inform you that your manuscript has been judged scientifically suitable for publication and will be formally accepted for publication once it meets all outstanding technical requirements.

Kind regards,

Chong Xu

Academic Editor

PLOS One
---

## [Editor Report · Acceptance letter]

PONE-D-25-21376R2

PLOS One

Dear Dr. Sato,

I'm pleased to inform you that your manuscript has been deemed suitable for publication in PLOS One. Congratulations! Your manuscript is now being handed over to our production team.

Kind regards,

on behalf of

Dr. Chong Xu

Academic Editor

PLOS One